# Mathematical Modeling and Pilot Test Validation of Nanoparticles Injection in Heavy Hydrocarbon Reservoirs

**Juan D. Valencia** [1,2], **Juan M. Mejía** [3,*] , **Matteo Icardi** [4] **and Richard Zabala** [5]

1 Facultad de Minas, Universidad Nacional de Colombia, Medellín Campus, Cr. 80 No. 65–223, Medellín 050034, Colombia; jdvalencl@unal.edu.co
2 Agencia Nacional de Hidrocarburos, Avenida Calle 26 No. 59–65, Piso 2, Bogotá 111321, Colombia
3 Facultad de Minas, Universidad Nacional de Colombia, Cr. 80 No. 65–223, Medellín 050001, Colombia
4 School of Mathematical Sciences, University of Nottingham, Nottingham NG7 2RD, UK; matteo.icardi@nottingham.ac.uk
5 Ecopetrol S.A., Cr. 13 No. 36–24, Bogotá 110311, Colombia; richard.zabala@ecopetrol.com.co
* Correspondence: jmmejiaca@unal.edu.co

**Abstract:** Heavy-oil mobility in reservoir rocks can be improved, using nanotechnology, by reducing the viscosity of the oil and improving the rock wettability to a water-wet condition. Previous pilot studies in Colombian heavy oil fields reported that nanoparticles dispersed in an oleic carrier fluid (diesel) increased oil production rates between 120–150% higher than before the interventions. However, to optimally deploy a massive nanofluid intervention campaign in heavy oil fields, it is valuable to implement simulation tools that can help to understand the role of operational parameters, to design the operations and to monitor the performance. The simulator must account for nanoparticle transport, transfer, and retention dynamics, as well as their impact on viscosity reduction and wettability restoration. In this paper, we developed and solved, numerically, a 3D mathematical model describing the multiphase flow and interaction of the nanoparticles with oil, brine, and rock surface, leading to viscosity reduction and wettability restoration. The model is based on a multiphase pseudo-compositional formulation, coupled with mass balance equations, of nanoparticles dispersed in water, nanoparticles dispersed in oil, and nanoparticles retained on the rock surface. We simulated a pilot test study of a nanofluid stimulation done in a Colombian heavy oil field. The injection, soaking, and production stages were simulated using a 3D single-well formulation of the mathematical model. The comparison of simulation results with the pilot test results shows that the model reproduced the field observations before and after the stimulation. Simulations showed that viscosity reduction during the post-stimulation period is strongly related to the detachment rate of nanoparticles. Simulation indicates that the recovery mechanism of the nanofluid stimulation is initially governed by viscosity reduction and wettability alteration. At latter times, wettability alteration is the main recovery mechanism. The nanoparticles transferred to the residual water promote the wettability alteration to a water wet condition. The model can be used to design field deployments of nanofluid interventions in heavy oil reservoirs.

**Keywords:** nanoparticles; modeling; simulation; field measurements; viscosity reduction

## 1. Introduction

Heavy oil is an important energy source. Approximately 40% of the world oil reserves are distributed in heavy and extra heavy oil reservoirs [1]. Density and viscosity of heavy oils (HO) are higher than light and intermediate oils, having a low mobility in the reservoir rocks. Asphaltenes and other heavy molecules are responsible for such behavior. The recovery factor of heavy oil fields is usually low. Thermal, chemical, and gas-based enhanced oil recovery (EOR) techniques have been used to increase the recovery factor of heavy oil fields. A review of EOR methods used in heavy oil reservoirs is presented by Guo et al. [2].

Chemical technologies, based on nanoparticles, have emerged as promissory options for enhanced/improved oil recovery applications [3–5]. Zabala et al. [6], injected a nanofluid containing alumina nanoparticles to inhibit asphaltene formation damage problems in a Colombian volatile field reservoir. The technique was extended to another reservoir having asphaltene problems, as reported by Franco et al. [7]. Zabala et al. [8] reported the use of nanofluids for stimulating 4 wells in two heavy oil fields in Colombia. A review of the application of nanotechnology in EOR applications is presented by Sun et al. [9], Afolabi [10], and Agi et al. [11].

Taborda et al. [12,13] showed that heavy oil viscosity can be reduced when engineered alumina and silica nanoparticles are dispersed in a carrier fluid. They suggested that the high surface energy of the tested nanoparticles adsorbed asphaltenes on the nanoparticles' surface and reduced the size of asphaltenes clusters. Consequently, a 52% reduction in heavy oil viscosity was measured by dispersing nanoparticles in a heavy oil sample. Later, Wang et al. [14] studied the microrheology of three heavy and extra-heavy oils, with $SiO_2$ nanoparticles at different concentrations. Time-domain nuclear magnetic resonance (NMR) and refractive indices were obtained to study the micro-rheological properties (transverse relaxation time and diffusion coefficients) of the dispersions. The study showed that there exists an optimum concentration of nanoparticles (around 1000 ppm) allowing a viscosity reduction between 35–45%. Moreover, they showed that the nanoparticles increase the NMR relaxation time and diffusion coefficient, indicating that the asphaltenes on the nanoparticles surface promotes the breakdown of aggregates. Patel et al. [15] found that metal oxides nanoparticles can reduce oil viscosity at temperatures lower than these of thermal EOR methods. They hypothesized that the Ostwald ripening is one of the mechanisms for viscosity reduction.

Liu et al. [16] studied the viscosity reduction in bitumen using nanoparticles ($TiO_2$ and CuO), and surfactants (ethyl cellulose and the quaternary ammonium salt of heptadecenyl hydroxyethyl imidazoline). They found a viscosity reduction between 39–43%, when pure nanoparticles were dispersed in the bitumen, while the reduction achieved with the surfactants was 19–32%. However, a synergistic effect was found when the nanoparticles and the solvents were simultaneously employed, reducing the bitumen viscosity in the range 70–79%. A similar work was performed by Montes et al. [17] who evaluated $SiO_2$ nanoparticles and 4 carrier fluids (xylene, diesel, *n*-pentane, and *n*-heptane using dimethylformamide as a nanoparticles dispersant). They found that the heavy oil viscosity, achieved with a concentration of nanoparticles of 1000 ppm in the carrier fluid, was 4% lower than that achieved with the carrier fluid alone. In addition, authors showed that the nanoparticles dispersed in the carrier fluid significantly increased the durability of viscosity reduction effect after 30 days of measurements.

Another interesting feature of nanoparticles is that the rock surface wettability can be altered from an oil-wet condition towards a water-wet condition. Maghzi et al. [18], Giraldo et al. [19], Li and Torsaeter [20], and Dahkaee et al. [21] reported a significant restoration of wettability from an oil-wet to a water-wet condition when alumina, silica, or NiO nanoparticles were used in their experiments. A similar result was found by Mohajeri et al. [22] using $ZrO_2$ nanoparticles. Sun et al. [9] summarizes the experimental studies related to wettability alteration using nanofluids for different oil/rock systems. According to Afolabi [10], deposited nanoparticles create an uneven charge distribution on the rock surface. One consequence is that water molecules are attracted since hydrogen bonds are created with the rock/nanofluid surface. On the other hand, when oil droplets are attached to the surface, nanoparticles increase the disjointing pressure because they can be located in their surrounding region. Afekare et al. [23,24] employed atomic force microscopy to investigate the oil release mechanisms when $SiO_2$ nanoparticles are injected in a clay-rich surfaces. The authors found a significant reduction in the adhesion force and adhesion energy: the wettability alteration was controlled by the reduction in non-electrostatic and interaction force and increasing of the electrostatic repulsion force.

Heavy oil mobility in the reservoir can be improved as a result of both the viscosity reduction and the increase in the effective permeability of heavy oil when nanoparticles are used in well stimulation operations and EOR operations. In the field tests presented by Zabala et al. [8] a silica-based nanofluid was injected into four heavy oil wells. The silica nanoparticles were dispersed in diesel, promoting a closer contact with heavy oil. They reported a satisfactory performance of the nanofluid, showing incremental production of 150%, on average.

The success of the massive implementation of nanofluids at field-scale for enhancing heavy oil mobility will depend on the understanding of the underlying physics taking place in the reservoir. A realistic model is required for designing, optimizing, and monitoring nanofluid-based field interventions. The model can be used to appraise the impact of nanoparticle concentration, injection rate, soaking time, among other operative variables, on the economic revenue.

Up to date, few mathematical models of multiphase flow, as well as nanoparticle transport and retention, have been published. El Amin, et al. [25–28] developed a mathematical model for studying different nanotechnology applications. They consider a two-phase flow model that is coupled to local mass balance equations for dispersed and retained nanoparticles. Particles are dispersed in liquid phases, and their retention on the rock surface is modeled using a critical velocity approach. Remobilization is also accounted for in the model formulation. They employ a numerical solution strategy based on IM-PEC (implicit pressure—explicit concentration). Concentration is solved using an iterative implicit scheme.

Mozo [29] presented a 1D, black-oil model (BOM) of the interaction of nanoparticles and heavy oil. Asphaltenes adsorption on the nanoparticles surface were assumed slow in the model formulation. The model was calibrated with experimental data. However, the model was computationally expensive, and numerical instabilities occurred during the simulations. The complexity of the model was reduced by Mozo et al. [30], by assuming a local adsorption equilibrium between the asphaltenes and nanoparticles, reducing numerical instabilities. The model presented by Mozo et al. [30] was successfully validated with two-phase flow core-flooding experiments injecting nanoparticles. Valencia et al. [31] extended the previous 1D black-oil model [30] to simulate the injection of nanoparticles in heavy oil reservoirs to a three-phase, 3D formulation, to simulate a field-scale case. Although the model presented by Valencia et al. [31] had good predictive capabilities, numerical simulations were slow, since the resulting set of equations is highly coupled. The mass balance equation of the asphaltenes, reported by Valencia et al. [31], imposed a highly coupling nature on the system of equations without improving the predictive capability.

In this paper, we present a new model for simulating well stimulations with nanofluids, which presents higher computational stability and reduces the complexity of the previous models, reported by Valencia et al. [31]. Here, we implemented a compositional formulation to simulate a BOM fluid model, with changes in the saturation pressure and fluid properties dependent on the composition. This BOM pseudo-compositional approach is coupled to the nanoparticle transport and retention equations. Here, we further simplified the previous models, eliminating the asphaltene transport equation. On the other hand, the numerical solution was improved by employing a preconditioned GMRES method (Generalized Minimal RESidual) [32] with incomplete Lower-upper (LU) Factorization (zero fill) algorithm to solve the resulting Jacobian. Simulations can be done in orthogonal grids (linear and radial) and non-orthogonal grids using a corner-point format. The model was implemented in the FlowTraM solver. The numerical implementation details are presented by Echavarría et al. [33]. The model was validated with core-flooding experiments and further verified with a pilot test study done in a heavy oil field in Colombia. The model presented in this work has better predictive capabilities and faster performance than the model presented by Valencia et al. [31]. The computational time was reduced by 25.9% in a 1D case and 78.2% in a 3D case.

The novelty of this research is the development of a model to simulate the use of nanoparticles improving the mobility of heavy oil at reservoir conditions. The model is capable of simulating the injection, soaking, and production stages of nanofluids-based stimulation deployments in oil fields, thus allowing for the design, optimization, and monitoring of these field interventions.

## 2. Mathematical Model

In this section, the governing and constitutive equations are presented.

### 2.1. Basic Assumptions

To formulate the mathematical model, the following assumptions are done:

- Negligible reservoir temperature variation in the nanofluid injection stage.
- Compositional changes, as a result of the injected nanofluid, are not relevant to the overall process performance.
- The carrier fluid (diesel and a nanoparticle dispersant) is miscible with oil at reservoir conditions, and the mixing between residual oil and diesel is instantaneous.
- Nanoparticles can be transferred between liquid phases, according to their hydrophilic/hydrophobic behavior.
- The nanoparticle attachment can be described using the double-site model from Zhang [34].

### 2.2. Governing Equations

The main components of the model are oil, gas, water, solid nanoparticles, and a carrier fluid (diesel and nanoparticles dispersant). These components are distributed in the following phases: oleic, volatile, aqueous, dispersed nanoparticles in oil, dispersed nanoparticles in water, and solid nanoparticles attached to the rock surface. Table 1 presents the distribution of phases/components.

**Table 1.** Distribution of components in phases, during nanofluids injection and production, in heavy-oil reservoirs.

| | | | Phases | | | |
|---|---|---|---|---|---|---|
| **Components** | **Oleic (o)** | **Volatile (g)** | **Aqueous (w)** | **NPs Dispersed in Oleic Phase (n-o)** | **NPs Dispersed in Aqueous Phase (n-w)** | **Attached Nps on Solid Matrix (n-s)** |
| Oil | x | | | | | |
| Gas | x | x | | | | |
| Water | | | x | | | |
| Carrier fluid (diesel and dispersant) | x | | | | | |
| Nanoparticles | | | | x | x | x |

The multiphase flow is modeled based on a compositional formulation, considering the pseudo-components gas, oil, and water from an extended black oil model. The Wang [35] approach is used to translate the PVT information to the compositional formulation. Equation (1) displays the general balance equation for each pseudo-component [36,37]:

$$\frac{\partial}{\partial t}(N_i) + \vec{\nabla} \cdot \sum_{p=1}^{n_p} (w_{i,p}\vec{u_p}) + \overline{q}_i = 0, \forall i \in \{1, 2, \ldots, n_i\} \tag{1}$$

where, $N_i$ is the $i$th pseudo-component moles, $w_{i,p}$ is molar fraction of $i$th pseudo-component in the phase $p$ (oleic, volatile, aqueous), $\boldsymbol{u}_p$ is the phase $p$ velocity, and $q$ is the well injection-production rate term.

The moles of $i$th component is defined as follows

$$N_i = \sum_p w_{i,p} W_p \varnothing S_p$$

where $S_p$ denotes saturation of phase $p$, and $W_p$ denotes total moles of phase $p$. Darcy equation is used to calculate phase velocity, by:

$$\vec{\boldsymbol{u}_p} = -\frac{\vec{\mathbf{K}} k_{r,p}}{\mu_p} \vec{\nabla} \Phi_p$$

where $\vec{\mathbf{K}}$ is the permeability, $k_{r,p}$ is the relative permeability of phase $p$, $\mu$ is the phase viscosity, and $\Phi$ is the phase potential.

Nanoparticles are injected, in diesel, into the reservoir, whereas we assume that complete and instantaneous miscibility occurs between oil and diesel. As a result, nanoparticles are emplaced in the oleic phase in the reservoir. In addition, some nanoparticles can be transferred to water. Dispersed nanoparticles in liquid phases can be attached onto the rock surface, according to the hydrophilic/hydrophobic nature of the nanoparticles and the rock wettability. Therefore, a model of the nanoparticle transport in the reservoir has to account for interfacial mass transfer between liquid phases, as well as attachment/detachment to/from the rock surface [38]. The molar balance equations, of the nanoparticles dispersed in the $p$ phase, are presented in Equation (2):

$$\frac{\partial}{\partial t}\left(\varnothing S_p \overline{\rho_p} c_{n,p}\right) + \vec{\nabla} \cdot \sum_{p=1}^{n_p}\left(\overline{\rho_p} c_{n,p} \vec{\boldsymbol{u}_p}\right) + \overline{\rho_p} c_{n,p} \overline{q}_p = \dot{N}_{n,p \to sj}, \; \forall i \in \{1, 2, \ldots, n_i\} \quad (2)$$

where $\varnothing$ is the porosity, $\overline{\rho_p}$ is molar density, $S_p$ is the phase saturation, $c_{n,p}$ is the molar concentration of nanoparticles $n$ in the phase $p$, and $\dot{N}$ is the nanoparticle molar transfer rate between phases. The subscript $n$ denotes nanoparticles, and subscripts $s_1$ and $s_2$ indicate solid matrix active sites 1 and 2, respectively, according to the two-site model [34]. This model was adopted to represent the attachment-detachment of nanoparticles, where site 1 represents the rock surface region, where nanoparticle attachment is irreversible, while site 2 represents the rock surface region where nanoparticle attachment is reversible. The irreversible attachment assumption of the two-site model [34] is consistent with many core-flooding experiments and observations from Afekare et al. [24] using atomic force microscopy.

The molar balance equation of deposited nanoparticles, on solid active sites 1 and 2, can be written as follows:

$$\frac{\partial}{\partial t}\left((1 - \varnothing)\overline{\rho_r} c_{n,sj}\right) = -\dot{N}_{n,o \to sj} - \dot{N}_{n,w \to sj}, \forall j \in \{1, 2\} \quad (3)$$

where $c_{n,sj}$ is nanoparticle coverage on rock active site 1 or 2.

### 2.3. Constitutive Equations

The attachment-detachment rate is modeled using the two active sites [34], where site 1 represents the rock surface region, where nanoparticle attachment is irreversible, while site 2 represents the rock surface region where nanoparticle attachment is reversible:

$$\dot{N}_{n,p \to s1} = K_{n,p \to s1}^{atr}\left(1 - \frac{c_{n,s1}}{c_{n,s1,max}}\right) c_{n,p} \overline{\rho_p} \varnothing S_p \quad (4)$$

$$\dot{N}_{n,p\to s2} = K_{n,p\to s2}^{atr}\left(1 - \frac{c_{n,s2}}{c_{n,s2,max}}\right)c_{n,p}\overline{\rho_p}\varnothing S_p - K_{n,s2\to p}^{rem}(1-\varnothing)\left(\frac{S_p}{S_L}\right)c_{n,s2}\overline{\rho_r} \qquad (5)$$

where $c_{n,s1,max}$ y $c_{n,s2,max}$ are the maximum molar coverage on active sites 1 and 2, $K_{n,p\to s1}^{atr}$ are the deposition rate parameter from phase $p$ to site 1, and $K_{n,s2\to p}^{rem}$ is the remobilization rate parameter from reversible active site 2 to the phase $p$. $S_p$ and $S_L$ are the phase $p$ saturation and the liquid saturation (oleic + aqueous). Equations (4) and (5) were modified from the ones presented by Zhang [34] to be consistent with the formulation of this model; the mass fraction coverage $(x_{n,sj})$ is related to the molar coverage by: $c_{n,sj} = x_{n,sj}M_r/M_n$, where $M$ is the molecular weight. The double site model parameters have to be obtained from experiments for nanofluids-reservoir and fluids-reservoir rock systems.

The nanoparticles' attachment and detachment on the rock surface change the available pore volume. A direct consequence is that porosity and permeability can be altered. Porosity changes when nanoparticles are attached on the reservoir, by:

$$\varnothing = \varnothing_0 - \delta\varnothing \qquad (6)$$

where, $\varnothing_0$ is the initial porosity, and $\delta\varnothing$ is the volume fraction of deposited nanoparticles:

$$\delta\varnothing = (c_{n,s_1} + c_{n,s_2})\varepsilon_r\frac{\overline{\rho_r}}{\overline{\rho_n}} \qquad (7)$$

where $\varepsilon_r$ is the rock volume fraction. Permeability also changes as a consequence of nanoparticle attachment. Permeability alteration is calculated following Civan et al. [39]:

$$K = K_0\left(\frac{\varnothing}{\varnothing_0}\right)^3 \qquad (8)$$

We model the wettability alteration through changes in the relative permeability curves before and after nanoparticle injection. Relative permeability curves are interpolated based on the available surface model created by the particle attachment [40]:

$$k_{rj} = k_{rj}^b + \frac{k_{rj}^a - k_{rj}^b}{a_{sp}}a_{tot} \qquad (9)$$

where $k_r^b$ and $k_r^a$ are, respectively, the relative permeability curves before and after the nanoparticle injection. $a_{sp}$ is the surface area per unit of volume of the porous rock [41]:

$$a_{sp} = 5051\varnothing\left(\frac{\varnothing}{K}\right)^{0.5} \qquad (10)$$

where $a_{tot}$ is the nanoparticle surface area per unit of volume:

$$a_{tot} = 6\frac{\beta}{d}\delta\varnothing \qquad (11)$$

where $\beta$ and $d$ are the particle sphericity and diameter, respectively. Based on the deposited nanoparticle concentration on active sites 1 and 2, the permeability and porosity reductions are updated at each iteration, as well as the relative permeability, for every cell of the computational domain.

For a given nanofluid-oil system, the oil viscosity $(\mu_o)$ dependence on the nanoparticle concentration can be written as follows:

$$\mu_o = \mu_o(P, T, c_{n,o}) \qquad (12)$$

where *P* and *T* are, respectively, the reservoir pressure and temperature. Here, viscosity measurements at different nanoparticle concentrations, pressures and temperatures are required. To this work, the viscosity measurements reported by Zabala et al. [8] were employed.

### 2.4. Numerical Solution

The model formulation yields a set of nine coupled partial differential equations. The partial differential equations were discretized using the finite volume method in Cartesian, cylindrical, and non-orthogonal grids. Spatial discretization considers structured grids and the variation of properties over computational grids. Permeability for each direction is also included in the numerical formulation. The time-domain was discretized using the "Euler backward" method. The resulting set of algebraic equations are non-linear and highly coupled. The non-linear system was solved using the Newton–Raphson method, following a fully coupled approach. The Jacobian was computed using a numerical scheme. The resulting set of linear equations was solved using preconditioned GMRES [32] with an incomplete LU Factorization (zero fill) algorithm. The numerical implementation details are presented by Echavarría et al. [33]. Primary variables are oil pressure, gas moles, water moles, dispersed nanoparticle concentration in water, dispersed nanoparticle concentration in oil, the carrier fluid concentration, attached nanoparticles on site 1, and attached nanoparticles on site 2. Saturations are further evaluated, as a secondary variable, by estimating the total moles of each phase, via K-values, and calculating their volume with the updated phase density at pressure *p*. The model was programmed using FORTRAN 2008. The numerical solution strategy is presented in Algorithm 1.

---

**Algorithm 1:** Single well reservoir modeling

**Data:** 3D static model, petrophisics, fluid model, operative conditions and process parameters
**Result:** Primary and secondary variables for the spatial discetization for each time step

1   initialization;
2   **while** $t < t_{Total}$ **do**
3     *Compute actual time properties* ;            `// Time n`
4     *Variable vector initializaion* ;    `// ` $X^k = X^n$`, Solution = False`
5     **while** *Solution* = *False* **do**
6       *Compute properties with* $X^k$;
7       *Equation system linearization* ;        `// Newton Method`
8       *Solve the linear system with preconditioned GMRES;*
9       **if** *Error < Tolerance* **then**
10         **if** *Mass Balance Criteria < Tolerance* **then**
11           *Solution = True;*
         **else**
12           *Refine the spatial/time domain and return to line 1;*
        **end**
      **else**
13         *Update variables* ;            `// ` $X^k = X^{k+1}$
      **end**
    **end**
14     *Update variables for the next time step;*      `// ` $X^n = X^{k+1}$
15      $t = t + Dt$
  **end**

---

## 3. Results and Analysis

In this section, the model performance will be assessed with core-flooding experiments. Then, we will compare measurements of a pilot test, done in an oil field in Colombia, with the simulation results.

### 3.1. Experimental Validation

The model was validated with experimental data from core-flooding setup from Zabala et al. [8]. Main properties and operational conditions are listed in Table 2, as well as the double-site model parameters. These parameters were estimated by Mozo et al. [30] from a core-flooding test of nanofluid injection and a durability test at reservoir conditions reported by Zabala et al. [8]; the parameters were estimated by matching the experimental observations of the oil recovery curve, before and after the injection, as well as the durability. Viscosity measurements for different nanoparticle concentrations are presented in Figure 1. The viscosity measurements are used as input to the simulation model and, then, interpolated to calculate the crude oil viscosity depending on the viscosity of the crude oil.

**Table 2.** Main fluid, operational, and model parameters used for the experimental verification.

| Parameter | Value |
|---|---|
| Length [cm] | 7 |
| Diameter [cm] | 3.75 |
| Permeability [mD] | 622 |
| Porosity [-] | 0.21 |
| Pore press. [bar] | 206.8 |
| Temp. [K] | 372 |
| Oil API | 8 |
| Initial pressure [bar] | 206.8 |
| Initial oil saturation [-] | 0.77 |
| Inj. rate [cc/min] | 0.6 |
| $K_{n,p \to s1}^{atr}[1/s]$ | $4.00 \times 10^{-4}$ |
| $K_{n,p \to s2}^{atr}[1/s]$ | $1.00 \times 10^{-4}$ |
| $K_{n,s2 \to p}^{rem}[1/s]$ | $4.00 \times 10^{-5}$ |
| $x_{n,s1,max}[g/g]$ | $1.06 \times 10^{-5}$ |
| $x_{n,s2,max}[g/g]$ | $2.02 \times 10^{-4}$ |
| Sphericity [-] | 0.9 |
| Nanoparticle diameter [nm] | 22 |

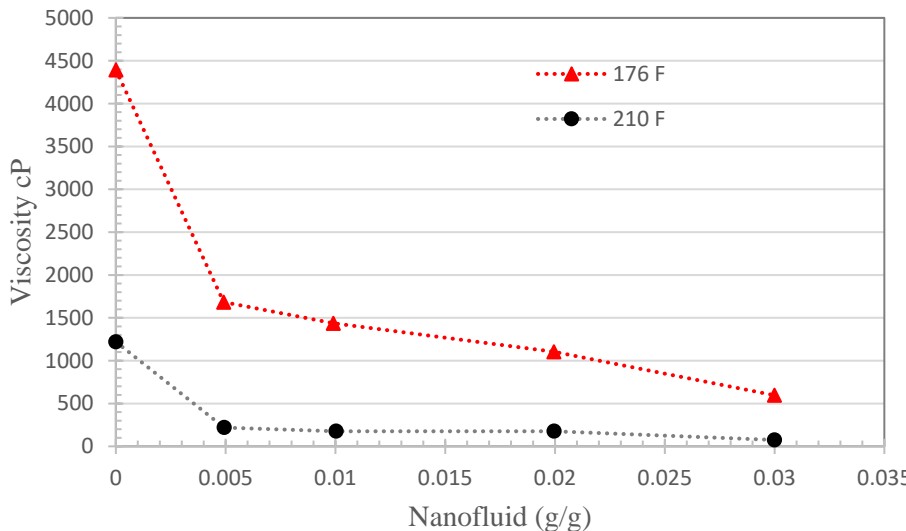

**Figure 1.** Viscosity measurements for different nanoparticle concentrations and temperatures (Adapted from Zabala et al., [8]).

The experimental protocol and conditions reported by Zabala et al. [8] were recreated in the simulation tool. Figure 2 reports the oil recovery curves, measured from the base case and after injecting 0.3 pore volumes of the nanofluid.

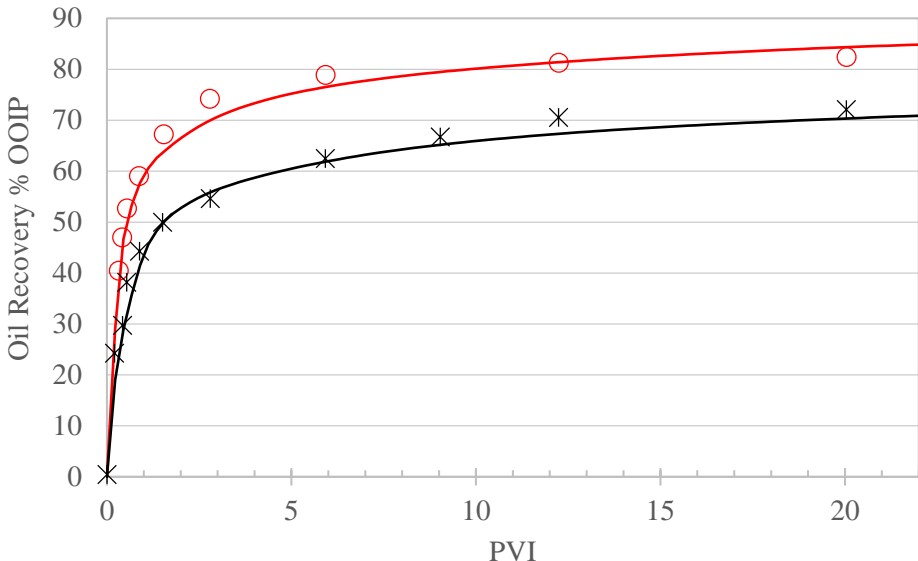

**Figure 2.** Oil recovery curve with nanoparticle injection (red set) and base case (black set). Symbols: experimental data (from Zabala et al., [8]). Continuous line: simulation results.

A good agreement between model results and experimental observations is noted, in Figure 2, for the base case and the nanofluid injection case. The root mean square error (RMS) of the base case recovery curve set is 3.4%, and the nanofluid injection one is 3.0%.

### 3.2. Reservoir Scale Results

Zabala et al. [8] reported the successful application of nanofluids in four wells of the Chichimene and Castilla fields (llanos Basin) in Colombia. The main characteristics of these fields are reported by Hartshorn [42]. Heavy oil is produced, in these fields, from multiple sandstone reservoirs of the Tertiary and Cretaceous periods. The sandstone matrix is oil-wet, and the main formation damage mechanism is due to drilling and workover operations (induced damage). The pilot was extended to other wells in these fields. Here, we present the results of the well denoted by C1. Main properties of the well C1 are listed in Table 3.

**Table 3.** Properties of well C1.

| Parameter | Value |
|---|---|
| Well radius [cm] | 20.7 |
| Average porosity [-] | 0.1081 |
| Average permeability [mD] | 953 |
| Reservoir pressure [bar] | 89.6 |
| Initial water saturation [-] | 0.20 |
| Carrier fluid | Diesel |
| Nanoparticle concentration [ppm] | 2000 |

The well C1 produces from formations denoted by A, A_50 and A_40. Each formation has two layers. A summary of the average porosity and permeability of the layers is shown in Figure 3.

A formation damage study showed that the main damage mechanism is due to organic deposits and emulsion formations in the near-wellbore area. The organic deposits are located in a radius of 0.23 m. Furthermore, induced formation damage by drilling and completion activities and damage associated with changes in relative permeability were also identified. The reservoir has an active water drive. The average reservoir pressure was correlated using information from an adjacent well.

The well stimulation using nanofluids was done on 3 October 2017. The field deployment had the following stages: a conditioning operation was done in order to stabilize fines

and minerals, injecting an aqueous batch. Next, the silica nanoparticles were dispersed in diesel at 2000 ppm. The intervention finished with a post-flush stabilizing stage. The well was shut-in for one day to let the nanofluid soak. Afterward, the well was opened for production.

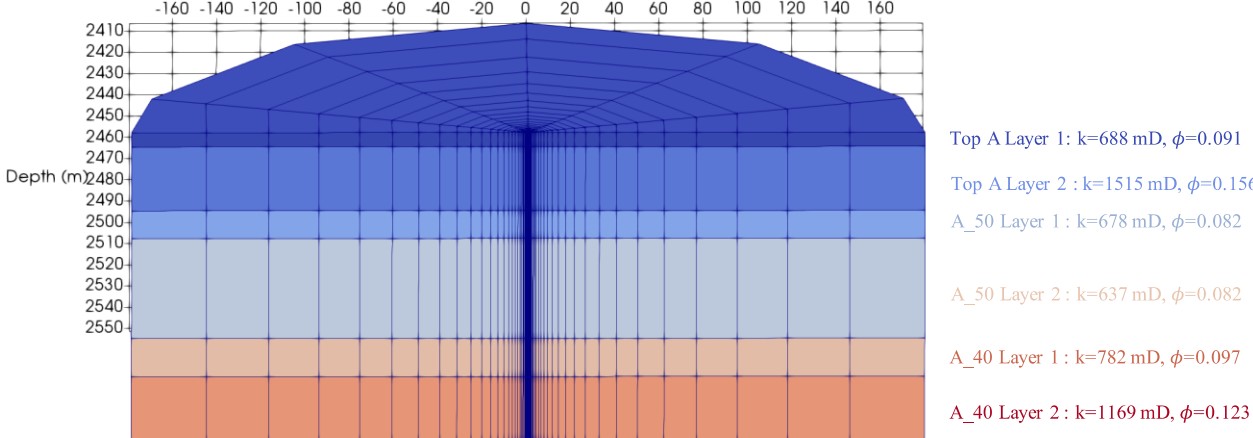

**Figure 3.** Main petrophysical properties.

The simulation grid was built on a single-well geometry. After a mesh size independence study, the grid has the following characteristics: six blocks were employed in the vertical direction (one block per layer), 10 uniform blocks in the angular direction, and 40 blocks logarithmically spaced in the radial direction. Model parameters were taken from Valencia et al. [31]. The skin factor of well C1, before and after the intervention, was history matched. The saturation of the oleic phase, before and after the stimulation, is presented in Figure 4.

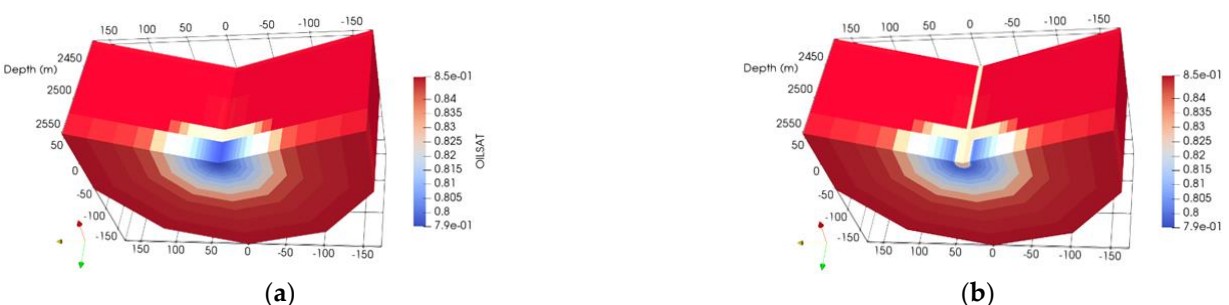

**Figure 4.** Oil saturation distribution, in a section of the reservoir, before (**a**) and after (**b**) the stimulation.

The oleic phase saturation in the nearby wellbore region, after the intervention, is slightly lower than that before the intervention because the nanofluid (nanoparticles and diesel as a carrier fluid) displaces the initial aqueous batch, as noted in Figure 4. Furthermore, the nanofluid also displaces oil, increasing its saturation further from the nanofluid injection front. Figure 5 presents the concentration of nanoparticles dispersed in oil.

Concentration of dispersed nanoparticles in oil after the stimulation reaches the injection concentration in the near-wellbore region, as noted in Figure 5. Concentration decays in deeper zones of the reservoir because of the mixing of nanoparticles with remaining oil and the attachment of nanoparticles on the rock surface (see Figures 6 and 7). After the soaking period, some nanoparticles are attached to the rock surface, reducing its concentration in the oleic phase. When the well is opened for production, the remaining nanoparticles that are dispersed in the oleic phase are flushed to the well. However, the concentration does not reach zero because nanoparticles are further detached from the rock surface when fresh oil comes from deeper, uncontacted zones of the reservoir. The

attached nanoparticle concentration distribution in rock active sites 1 and 2 are displayed in Figures 6 and 7, respectively.

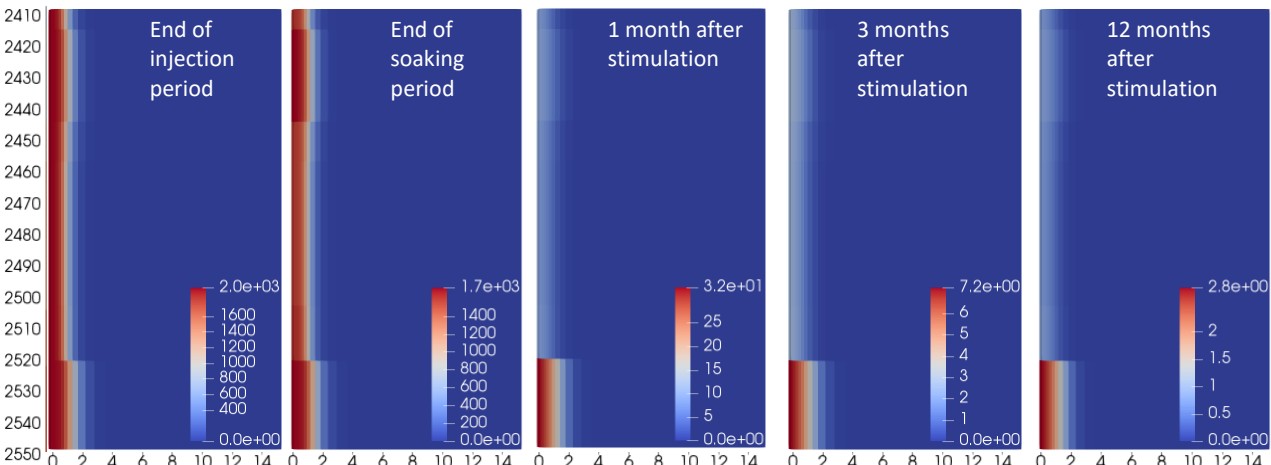

**Figure 5.** Nanoparticle concentration in oil (units in ppm) at different times. (2D view: plane $r,z$ at $\theta = 0$).

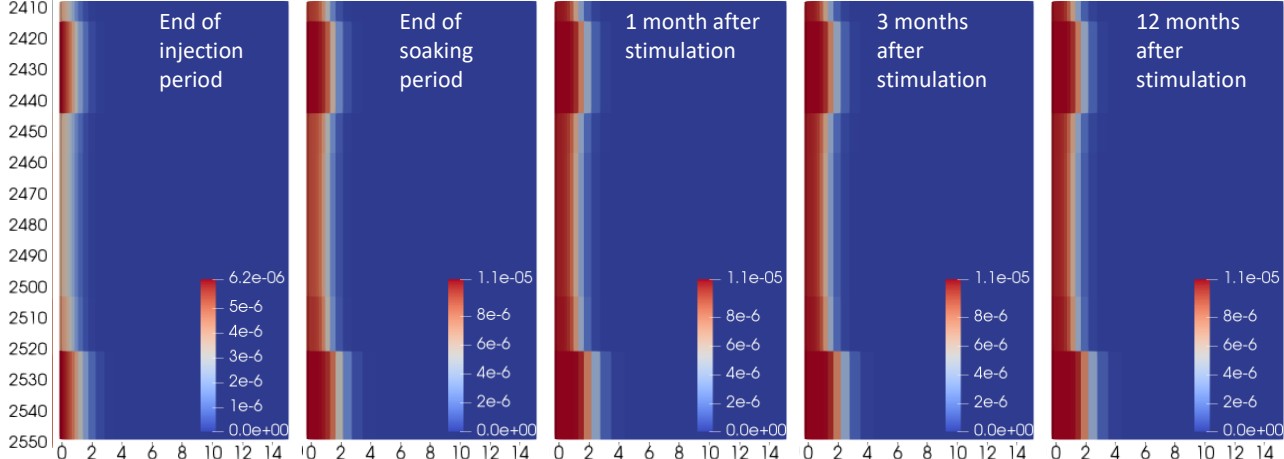

**Figure 6.** Nanoparticles retained on rock active site 1 (units in $w/w$) at different times. (2D view: plane $r,z$ at $\theta = 0$).

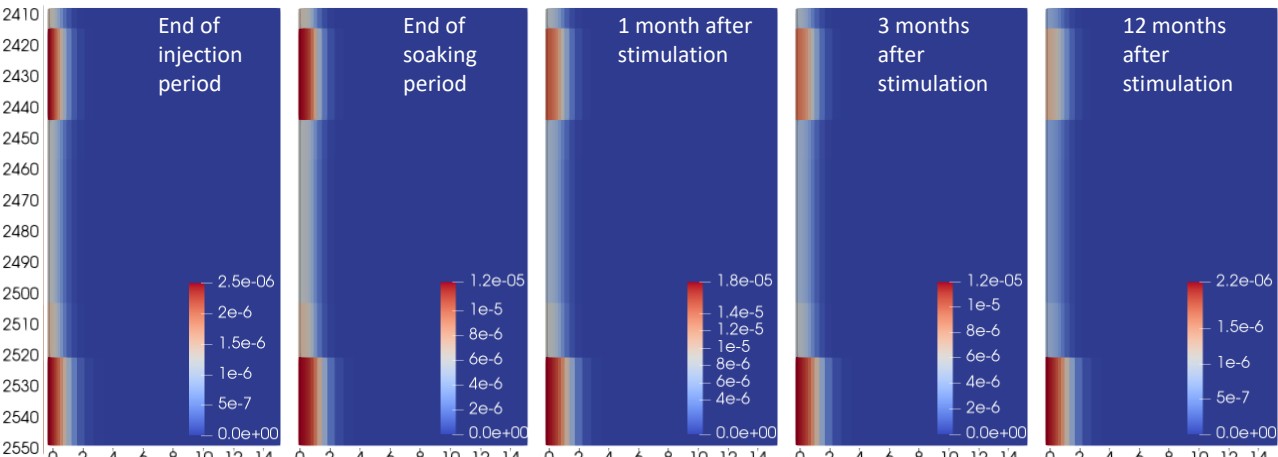

**Figure 7.** Nanoparticles retained on rock active site 2 (units in $w/w$) at different times. (2D view: plane $r,z$ at $\theta = 0$).

Figure 6 shows that the deposited particle concentration in active site 1 increases during the soaking and early production periods. The latter is a result from the mass transfer of nanoparticles from the oleic phase (see Figure 5) to the rock surface. The nanoparticle distribution in active site 1 remains constant at later times, staying consistent with the site 1 irreversible assumption stated by the two active sites model [34]. After one year from the intervention date, it is noted, in Figure 6, that the retention site 1 is saturated between the first 1–2 m of the reservoir, depending on the reservoir layer.

On the other hand, site 2 follows a different dynamic behavior from site 1. During the injection, soaking stages, and early days of the production stage, nanoparticle concentration on active site 2 increases since the concentrations of dispersed nanoparticles in oil and water are high enough to be transferred to the matrix (see Figure 7). At later times, dispersed nanoparticles in liquid phases are transported to the wellbore. When oil and water from deeper zones flow through the invaded area of the treatment, nanoparticles are released from active site 2 and transferred to the liquid phases, according to Equation (4). Nanoparticle concentration on active site 2 is reduced, as noted in Figure 7, providing nanoparticles to oil and water (see Figure 5). The nanoparticle detachment to the oil phase further reduces its viscosity. The viscosity distribution in the near-wellbore region is shown in Figure 8.

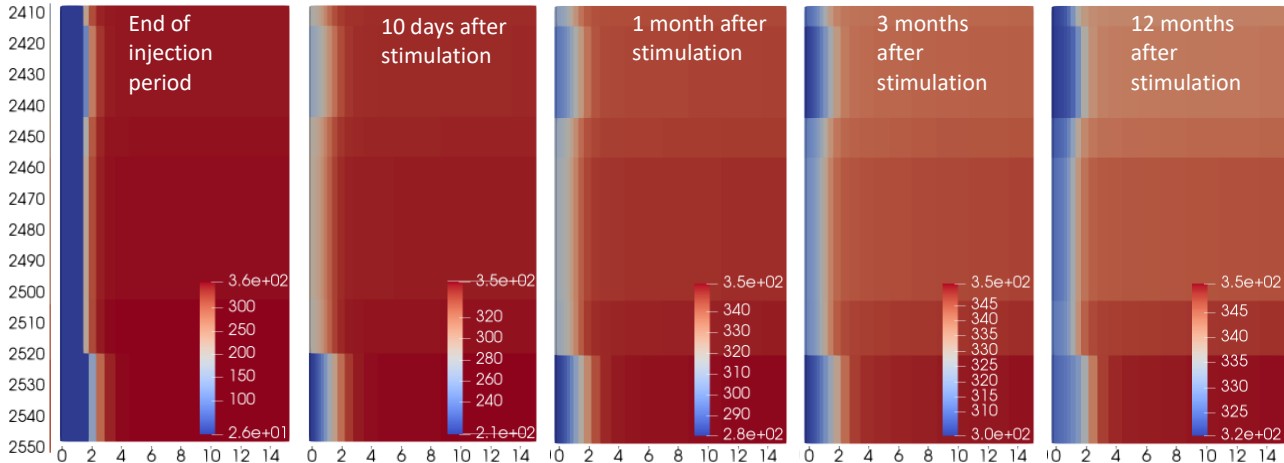

**Figure 8.** Oleic phase viscosity (units in cp) at different times. (2D view: plane *r,z* at *θ* = 0).

Viscosity in the near-wellbore area, after the soaking period, is dominated by the nanofluid viscosity. Once the well is opened for production and the remaining nanofluid is produced, remobilized nanoparticles in oil will decrease the phase viscosity, as observed in Figure 8. However, when the nanoparticles concentration in oil is reduced, the viscosity in the near-wellbore region will increase, as noted after 3 months, from stimulation (see Figure 8). The viscosity reduction is only impacted in the early post-production stage. Figure 8 shows that the oil viscosity distribution does not change significantly after the third month.

Measured and simulated oil production, before and after the nanofluids injection, is presented in Figure 9.

After the intervention, the simulation results, presented in Figure 9, show a short pulse in the oil production rate as a result of the injected nanofluid and some mobilized oil. A significant increase in the oil production rate occurs after the first peak, having a maximum value of 40 m$^3$/day. The dispersed nanoparticles in the oleic phase reduce the phase viscosity (see Figure 8) and, consequently, increase the flow rate. The trend of the measured oil production peak, after the post-stimulation period, was correctly captured by the mathematical model, as noted in Figure 9, between the stimulation date and November 2017. During this period, the nanoparticles attached to the rock surface, as theoretically predicted in Figures 6 and 7, promote a production increase due to the

wettability restoration. In addition, there is a marked reduction in oil viscosity (see Figure 8) in the early post-production period, which also enhances heavy oil mobility in the near-wellbore region. The observed behavior in oil production suggests that the viscosity reduction is an important mechanism during the first months. The peak is followed by a decline in the production rate, as observed in Figure 9. The oil decline rate is lower than that before the well intervention, suggesting that nanoparticles are impacting the oil mobility at a lower extent than the previous period. Detached nanoparticles generate a reduction in viscosity of only 8% in the near-wellbore area. Furthermore, attached nanoparticles on site 1 reach the maximum (saturation) concentration after three months, which suggests that the wettability restoration is the main mechanism in this period. The model predicts that the nanoparticles attached to rock active site 2 are remobilized to liquid phases, thus acting on the viscosity reduction mechanism, as well as wettability alteration, at early post-production stages. On the other hand, nanoparticles deposited on active site 1 remain over time, acting on the wettability alteration mechanism. However, it is not clear how nanoparticles deposited on the rock surface are deactivated. More research is needed to clarify this point. The overall behavior of the oil production rate was correctly captured by the model. The RMS error is 4.7 m$^3$/d.

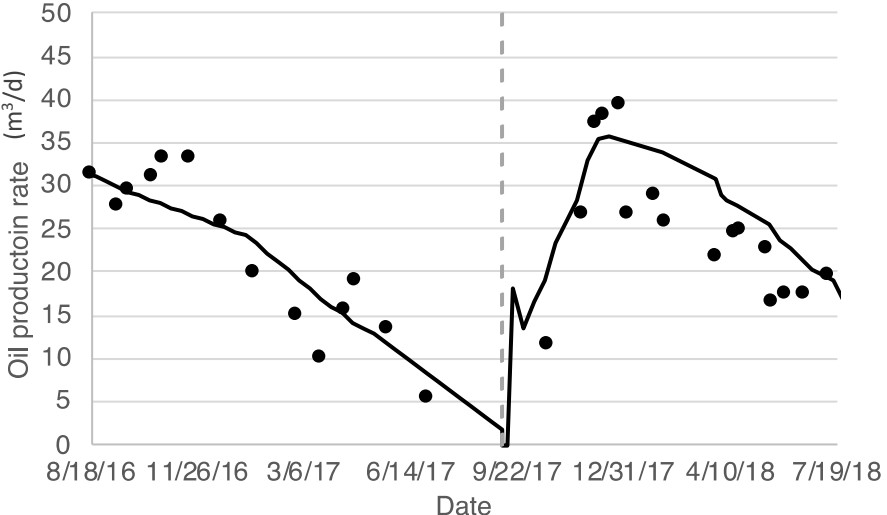

**Figure 9.** Measured and predicted oil production of well C1. Symbols: measurements. Continuous line: simulation results. The vertical dashed line denotes the beginning of the well stimulation. The red line denotes the oil decline.

Figure 10 presents the evolution of the water cut before and after stimulation.

It is observed in Figure 10 that there is a 9% increase in the water cut (see Zabala et al. [8]). Taking into account that the water drive mechanism is acting on the reservoir, the increase in mobility propagates the pressure front to deeper zones in the reservoirs. As a result, the water influx increases. Although the model slightly overestimates the water cut before the intervention, the increase trend is well captured. After the intervention, there is an increase in the water cut slope, which is correctly captured by the model. The RMS error is 0.11. The dispersed nanoparticle concentration in the production water is presented in Figure 10.

The maximum nanoparticle concentration in the aqueous phase occurs after the well is opened for production, as noted in Figure 11. The rock surface and oil phase, in the near-wellbore region, have the maximum nanoparticle concentration. Therefore, the nanoparticle mass transfer rate to the aqueous phase is fast because the potential gradient is high. Afterwards, simulated nanoparticle concentration is gradually reduced to 23 ppm at the end of 2017 and, later, to 2.6 pm at the end of 2018 (not shown here). Simulation results, presented in Figure 11, estimate lower nanoparticle concentration in the aqueous phase. However, the overall trend is correctly captured. The RMS error is 217.6 ppm.

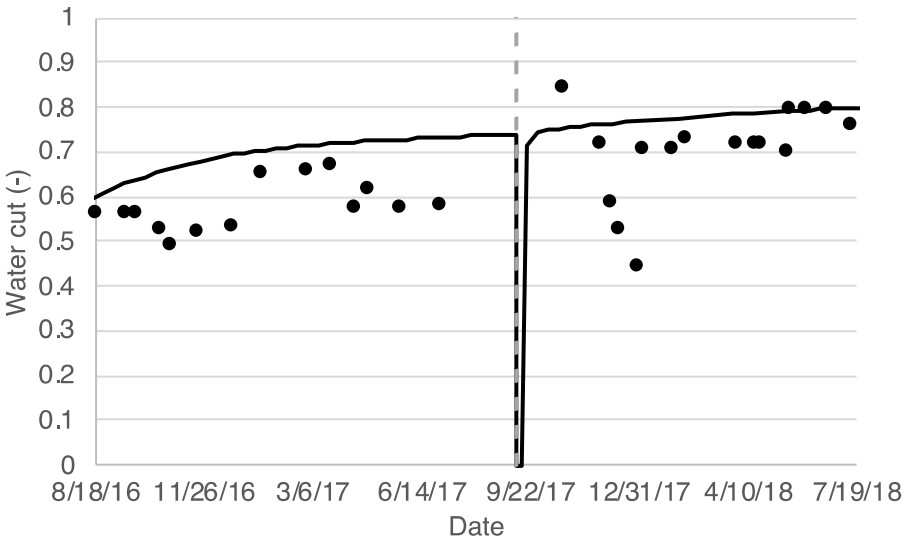

**Figure 10.** Measured and predicted water cut of well C1. Symbols: measurements. Continuous line: simulation results. The vertical dashed line denotes the beginning of the well stimulation.

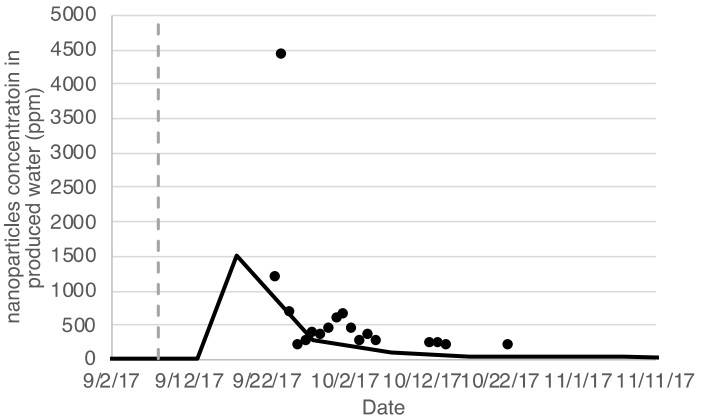

**Figure 11.** Measured and predicted nanoparticle concentration in the produced water of well C1. Symbols: measurements. Continuous line: simulation results. The vertical dashed line denotes the beginning of the well stimulation.

## 4. Summary and Conclusions

In this work, we presented a 3D numerical model that captures most of the transport and retention/remobilization mechanisms occurring when injecting nanoparticles in heavy oil reservoirs. The model is composed of nine coupled partial differential equations derived from mass conservation equations for oil, gas, water, asphaltenes, and nanoparticles. In addition, the model is formulated to account for non-equilibrium nanoparticle attachment and detachment from the rock surface. The model, presented here, was discretized using the finite-volume method and solved employing the Newton method. The numerical method was improved by preconditioning the Jacobian matrix using incomplete LU factorization.

A core flooding measurements of oil recovery showed a close agreement with the simulation results. In addition, the measured effective oil permeability, measured during the durability test, was correctly predicted by the model. The latter suggests that the mathematical model presented in this work captures most of the transport, transfer, and surface phenomena occurring when injecting nanoparticles that change the wettability of the system and reduce the viscosity of the heavy oil.

A pilot test-well intervention in a Colombian heavy oil reservoir was simulated using the developed model. The model correctly predicted the oil production rate, water rate, and produced nanoparticles after the intervention. The simulation results were in close

agreement with field measurements. The pilot study and simulation results showed that the nanoparticle attachment on the rock surface plays an important role for the deployment optimization: nanoparticles being partitioned from the oil-based treatment to the residual water during the injection stage are the ones acting on the wettability alteration.

The overall effect of nanoparticles changing the oil mobility in the reservoir follows a dynamic interaction between nanoparticles in oil, deposited nanoparticle on the rock surface, and fluid flow behavior during the injection and production stages. Nanoparticles residing in oil reduce the viscosity, mainly, during the early production period. Then, these nanoparticles are transported to the wellbore, and the viscosity reduction effect is reduced. The nanoparticles attached to the rock surface keep changing wettability, and the durability is gradually reduced as nanoparticles are detached from the rock surface and transported to the well, as shown by measurements and simulations. The particles transferred to the oil phase adsorb asphaltenes and reduce viscosity, and eventually, they can change the surface wettability towards an oil-wet condition if they are re-attached to the rock surface. A direct consequence of the asphaltenes' adsorption on the nanoparticles is a reduction in the transfer rate to the water phase.

Therefore, the incremental oil rate and the durability can be optimized, using the model, to maximize the economics of the intervention.

**Author Contributions:** Conceptualization, J.M.M., J.D.V. and M.I.; Methodology, R.Z. and J.M.M.; software, J.D.V.; validation, J.D.V., J.M.M. and J.D.V.; investigation, J.M.M.; resources, J.M.M. and R.Z.; data curation, J.D.V.; writing—original draft preparation, J.M.M.; writing—review and editing, J.M.M. and J.D.V.; visualization, J.M.M. and J.D.V.; supervision, J.M.M.; project administration, J.M.M. and M.I.; funding acquisition, J.M.M. and M.I. All authors have read and agreed to the published version of the manuscript.

**Funding:** This research was funded by the Royal Academy of Engineering—RAENG, grant numbers IAPP18-19\285, and TSP2021\100342; Fondo nacional de financiamiento para la ciencia, la tecnología y la innovación Francisco José de Caldas—MINCIENCIAS, and the Agencia Nacional de Hidrocarburos for financial support under contracts no. 272-2017 and 064-2018.

**Institutional Review Board Statement:** Not applicable.

**Informed Consent Statement:** Not aplicable.

**Acknowledgments:** Authors thank to the Royal Academy of Engineering, MINCIENCIAS, and the Agencia Nacional de Hidrocarburos, for financial support.

**Conflicts of Interest:** The authors declare no conflict of interest.

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
