# Peer review of "Mathematical Modeling and Pilot Test Validation of Nanoparticles Injection in Heavy Hydrocarbon Reservoirs"

_fluids, doi:10.3390/fluids7040135_

Round 1

Reviewer 1 Report

Comments attached

Author Response

First of all, we would like to thank the reviewer for the useful comments and questions on this manuscript, leading to improve quality of the article.   

  1. Line 12-13, Page 1: The author states that nanotechnology can restore rock wettability to water- wet condition but, in the reviewer’s opinion, this statement is somewhat misleading. The initial wettability of heavy oil reservoirs is not always water-wet because it depends on different factors such as saturation history, formation of hydrocarbons and impact of crude oil composition among others. Please revise.

Response 1: We agree with the reviewer. The sentence was modified.

  1. I am not sure if it is appropriate to cite previous works in an abstract and I’ll kindly advise that the authors check the journal format and abstract writing style.

Response 2: The citation in the abstract has been removed. The sentence was modified and adapted.

  1. Lines 16-17, Page 1: the author’s claim for the need of a reservoir simulator to deploy massive intervention campaigns can be strengthened by a clearer statement.

Response 3:  The statement was strengthened by clarifying the need of a reservoir simulator.

  1. Lines 17-18, Page 1: what does “nanoparticles dynamics” mean? Please clarify.

Response 4: “nanoparticles dynamics” was clarified

  1. Lines 60-61, Page 2: what type of oil can be used as dispersants based on the referenced work?

Response 5:  Different carrier fluids can be used, according to the literature. The carrier fluid must have a dispersant for stabilizing the nanoparticles. In the field test, diesel was used as a carrier fluid along with a dispersant.

  1. Line 63, Page 2: what do microrheological properties? A brief definition will be helpful.

Response 6:  Transverse relaxation time and diffusion coefficient properties are used to analyse the stress-strain at the microscale. A comment was included in the text

  1. Line 67, Page 2: MNR should be corrected to NMR.

Response 7: The typo was corrected

  1. Line 89, Page 2: significatively should be changed to significantly. Also, the word “perdurability” may be replaced with a more common word recognized in the petroleum engineering field.

Response 8: significatively was changed to significantly.  Durability is used instead of perdurability.

  1. Lines 100-102, Page 4: That the author reviewed a fundamental mechanism of nanoparticle injection which is disjoining pressure is laudable. However, it is also important to review recent and relevant literature. Recently, reduction in adhesion force and energy has also been identified as scientific oil release mechanisms in the case of silica nanoparticles (see Afekare et al. 2020, 2021 below)

Response 9: We updated the paragraph, with these findings. 

  1. Lines 107-109, Page 4: what’s the technical and commercial feasibility of injecting diesel-dispersed nanoparticles when diesel is a valuable petroleum product? Please comment on this, given that a related study was reviewed.

Response 10:  Diesel-based treatments are common in the oil and gas industry when an oil-based solution must be pumped into the reservoir. Diesel must be inhibited to avoid wettability problems, by using alcohols and surfactants. In addition, diesel is much cheaper than other organic compounds.  In most cases, the results are economically viable when considering the improvement in production.

  1. Line 114, Page 4: the article “the” between impact and nanoparticles should be replaced with “of”.

Response 11: The error was corrected.

  1. Line 132, Page 5: since the author stated that Valencia et al. (2019) extended their model, it is important that this previous model is briefly reviewed.

Response 12:  Agree. The model refers to the one presented by Mozo et al. (2018). A comment was included.

  1. Line 139-140, Page 5: the author needs to clearly define what “predictive capabilities” and “a faster performance” means. The article needs to clearly state the novelty of this work and it is distinguished from previous studies.

Response 13: The description of the model was improved. The novelty is highlighted, and it is clearly differenced from previous works.

  1. Line 141, Page 5: what does “pseudo-compositional formulation” mean?

Response 14: The pseudo-compositional formulation refers to a compositional approach to simulate a black-oil fluid model. Here, K-values are used instead of the solution gas-oil ratio (Rs), solution gas-water ratio (Rw), and oil-gas solution ratio (Rg).

  1. Line 143, Page 5: define LU in full.

Response 15: LU was defined.

  1. Line 149, Page 5: the author’s characterization of the proposed model as “simple but robust” is somewhat ambiguous. Please present a clearer statement.

Response 16:  The sentence was deleted.

  1. Line 164, Page 6: I was under the impression that diesel was used as dispersant - which fluid is being referred to as “nanoparticles dispersant” here?

Response 17:  The carrier fluid is composed of diesel and dispersant. A dispersant must be used to keep the particles stable and dispersed in the fluid.  The dispersant and the nanoparticles were provided by the company Petroraza. The dispersant composition is unknown by the authors.

  1. Line 198, Page 7: What type of silica nanoparticles was modeled in this work that makes it present hydrophilic and hydrophobic parts? Is there an underlying assumption that the particles were functionalized with some chemicals? Please clarify. Please also comment on the impact of rock wettability on attachment of the nanoparticles.

Response 18:  The authors do not know the extent of the hydrophilic and hydrophobic parts of the nanoparticles.  The authors do not know the details of the synthesis and functionalization of the nanoparticles.

For engineering purposes, since the purpose of this work is to model the process, here it is assumed that the nanoparticles are amphiphilic, so they can be partitioned into oil and water.  The model considers this aspect. Please refer to Eq. (2).

  1. Line 208-209, Page 8: what is the physical meaning of the two solid matrix sites? In other words why was this model adopted?

Response 19: This model was adopted to represent the attachment - detachment of nanoparticles, where site 1 represents the rock surface region where nanoparticle attachment is irreversible while site 2 represents the rock surface region where nanoparticle attachment is reversible. The selection of this model relies on experimental observations from Wang (2012) who showed that some nanoparticles are irreversibly attached to the matrix surface, while others can be detached by changing operating conditions (shear rate, salinity, concentration, etc). This behavior is consistent with experimental observation, including the ones reported by Afekare et al. (2021). In addition, the two-sites model matched most of the experimental measurements reported by Wang (2012) using silica nanoparticles. A comment was included in the paragraph

  1. Line 246, Page 9: add “model” after “surface”

Response 20: Done

  1. Line 268-269, Page 9: How do the viscosity measurements presented in Zabala et al.’s work relevant to this study? Please briefly clarify.

Response 21: viscosity measurements at different nanoparticles concentrations are required. To this work, the viscosity measurements reported by Zabala et al. (2016) were employed. The latter was included in the manuscript.

  1. Line 275-276, Page 9: What was the rationale behind choosing these different grid types?

Response 22: Different grids are required to simulate different systems. For instance, a core-flooding requires a cartesian grid, the well-intervention demands a radial geometry, and a non-orthogonal mesh is needed in a sector/full-scale model to account for reservoir structure/features.

  1. Figure 2: More distinct colors and symbols are recommended for clarity of presentation, and the flow rate and pressure drop profiles are also required.

Response 23: Done

  1. Figure 3: Were the petrophysical parameters and reservoir layering selected based on field data such as well logs, well test, special core analyses etc.? Please clarify

Response 24: The petrophysical data was provided by Ecopetrol S.A., from wells logs, well testing, and CCAL.

  1. Figure 5: If the nanoparticle concentration in the oil phase decreases over time during stimulation, will that not have an adverse effect on the oil production profile? Please comment on this

Response 25: On the contrary, it is expected that during the injection period, some of the nanoparticles in oil will be attached to the rock surface and promoting the wettability alteration. When the well is opened for production, some of the retained nanoparticles will be dragged to the oil, reducing the phase viscosity.

  1. Figures 6 & 7: The author needs to explain the reason behind the classification of the reservoir pay zone using two sites

Response 26:  The concentration distribution along the different pays is a consequence of the vertical heterogeneity of the reservoir. More conductive plays allow for a deeper nanoparticles emplacement. During the production stage, nanoparticles from site 1 are mobilized to the oil phase, reducing the concentration by about one order of magnitude after one year. Some of the detached nanoparticles from site 1 are further re-attached to the matrix active site 2, as indicated in Fig 7.

  1. Lines 410-415, Page 18: Why does an increase in nanoparticle concentration in the oil phase decrease oil viscosity? Please explain

Response 27:  Experimental measurements of oil viscosity presented in Zabala et al. (2016)-Figure 5, indicates that viscosity is reduced when the nanoparticles concentration increases, as follows:

The viscosity reduction is captured by the model.

  1. Lines 429-430, Page 18: That the author mentioned wettability alteration has a mechanism of oil production increase is not clear. Please provide supporting evidence.
  2. Lines 432-433, Page 19: Please see comment 28

Response 28 & 29:  Core flooding experiments and the pilot studies results demonstrated that the operation has long-lasting durability.  The model results captured this important observation.  At later times, there are particles that are irreversibly adsorbed on the matrix surface (active site 1). It is expected that these nanoparticles alter the surface wettability towards a more water-wet condition. Wettability restoration to a water wet condition is one of the enhanced/improved oil recovery mechanisms. The rationale behind it is that productivity increases as a result of reducing the energy required for the fluid flow.

  1. Figure 10: Change caption from “Figure 910” to Figure 10

Response 30: The typo was corrected

  1. Lines 444-445: The statement on the comparison of water cut profile presented in Figure 10 with previously published works is incomplete and somewhat misleading. How does the “other wells” behave in terms of water cut? Please clarify

Response 31: Agree. The statement was deleted.

In the pilot campaign, four wells were stimulated with the nanofluid. Three wells showed a reduction in the water cut, but the well analyzed in this work showed an increase in the water cut.

  1. Figure 11 & Lines 457-463: The author should explain how the nanoparticle concentration profile were obtained in the field. Which test was used, what was the procedure? Please cite relevant literature as well.

Response 32:  During the production stage, multiple separator tests were performed to monitor the oil and water production rates. Water samples were collected for further analysis, including nanoparticles concentration by a third-party company. The authors do not know the test and procedure, as long as the focus of this work is to model the operation.

  1. Lines 467-468: In the reviewer’s view, the author’s statement on the physics associated with nanoparticle injection in heavy oil reservoir is somewhat misleading. There is no consensus in the EOR community on underlying mechanisms of nanoparticle injection in heavy oils so it may not be stated that most of the physics have been captured in this work. Please rephrase.

Response 33: We agree. The sentence was rephrased

References

Afekare, D.; Garno, J.C.; Rao, D. Insights into Nanoscale Wettability Effects of Low Salinity and Nanofluid Enhanced Oil Recovery Techniques. Energies 2020, 13, 4443.

https://doi.org/10.3390/en13174443

Dayo Afekare, Jayne Garno, Dandina Rao, Enhancing oil recovery using silica nanoparticles: Nanoscale wettability alteration effects and implications for shale oil recovery, Journal of Petroleum Science and Engineering, Volume 203, 2021, 108897, ISSN 0920-4105,

https://doi.org/10.1016/j.petrol.2021.108897

Reviewer 2 Report

Manuscript ID: fluids-1338704 Mathematical modeling and experimental verification of nanoparticles

injection in heavy hydrocarbon reservoirs for wettability alteration

the paper deals mainly with the usage of nanoparticles to improve heavy oil mobility. 3D mathematical models were developed in the presented manuscript. The manuscript is well structured and written. However, a few gaps in the model development and numerical results need to be addressed properly. The manuscript may be suitable to be published with a 'major revision.'

Comments:

  1. The title is confusing as it contained an "experimental verification" while no experimental was carried out in the study. Moreover, the authors don't differentiate between "verification" and "validation" in modelling. Furthermore, most of the paper focuses on viscosity and barely touches the wettability.
  2. (Line 14) The abstract should focus on your original research, not on the work of others. Avoid citing sources in your abstract.
  3. Introduction covered a wide range of previous related works, which is great for researchers and engineers not related to this field. In addition, as this is research, not a review paper, having five pages addressed the literature review is not necessary. Focusing on the most recent and related previous work is essential.
  4. (Line 190 & 203) Equations 1 and 2 are missing references.
  5. (Line 227 & 228) what kind of modification was implemented in equations (4) and (5) to align the need of this study?
  6. (Line 240) equation 7 has no reference, and please define all the parameters
  7. (Line 275 & 276) "The partial differential equations were discretized using finite volume method in cartesian, cylindrical and non-orthogonal grids." There were no results presented reflect what the authors have done.
  8. There are some typos, e.g., (Line 287) "secondary" instead of "secondary" in the algorithm.
  9. There were no experiments done in this study. Not sure why there is a whole section of it? It is usually common for each developed mathematical model, and it needs to be verified and then validated. What in this section, to my best knowledge, is really validation, not verification.
  10. (Line 300 & 301) the author mentioned they used a history matching of recovery factor. The authors need to address and explain this.
  11. Most, if not all, of the figures, are lack explanation and interpretation. For example, the authors just put figure 1 from the previous study without giving details of the reason behind it and why they address it here. They apply to all figures and charts.
  12. (Figure 2) instead of having a long figure title, it would be much easier if there is a legend.
  13. (Line 351) "The skin factor of well C1, before and after the intervention, was history matched." Please explain.
  14. (Line 397) "site 2 follows a different dynamic behaviour from site 1." What kind of different dynamic do the authors refer to, and why does this behaviour occur?
  15. Overall results section, this section has great results and promising work. However, it lacks interpretation and explanation of these results. For example, (Line (430)) "Then, a decline in the production rate is observed in Figure 9."
  16. Some references needed adjustment and corrections. Would you please review them carefully?

Author Response

Frist of all, we’d like to thank the reviewer for the comments and suggestions to improve the quality of this paper.

Comments:

  1. The title is confusing as it contained an "experimental verification" while no experimental was carried out in the study. Moreover, the authors don't differentiate between "verification" and "validation" in modelling. Furthermore, most of the paper focuses on viscosity and barely touches the wettability.

Response 1: We agree with the reviewer, the title was rewritten. The new title is:

“Mathematical modeling and pilot test validation of nanoparticles injection in heavy hydrocarbon reservoirs”

  1. (Line 14) The abstract should focus on your original research, not on the work of others. Avoid citing sources in your abstract.

Response 2:  The citation in the abstract has been removed. The background of nanoparticle implementation in Colombia was left in a general way and specific information related to the previous work was deleted.

  1. Introduction covered a wide range of previous related works, which is great for researchers and engineers not related to this field. In addition, as this is research, not a review paper, having five pages addressed the literature review is not necessary. Focusing on the most recent and related previous work is essential.

Response 3: Some unneeded references were removed.

  1. (Line 190 & 203) Equations 1 and 2 are missing references.

Response 4: We do not agree with the reviewer. Equation 1 is a mass balance equation for the ith component. Equations 2 is a mass balance equation for the nanoparticles.

  1. (Line 227 & 228) what kind of modification was implemented in equations (4) and (5) to align the need of this study?

Response 5:  Zhang (2012) developed the model for single-phase and mass-based approaches. In this sense, Transfer is made dependent on the amount of nanoparticles in each phase by considering the phase saturation (in that sense for fluid saturations close to 1 the model is similar to the one proposed by Zhang, while for saturations close to zero the nanoparticle transfer from that phase disappears).

  1. (Line 240) equation 7 has no reference, and please define all the parameters

Response 6: Equation 7 is a volume balance, developed by the authors, representing the volume occupied by the nanoparticles retained on sites 1 & 2.  The missing parameters were included in the paper (ε_r is the rock volume fraction)

  1. (Line 275 & 276) "The partial differential equations were discretized using finite volume method in cartesian, cylindrical and non-orthogonal grids." There were no results presented reflect what the authors have done.

Response 7: We do not agree with the reviewer.

Core-scale simulations were done in a cartesian grid in section 3.1. The single-well simulations were done in cylyindrical coordinates in section 3.2 .  For the sake of generality, the model also accounts for a non-orthogonal grid as stated in case the model is extended to a sector of the reservoir or to the overall extent of the reservoir. The implementation and validation of the approach including non-orthogonal meshes are presented in Echavarría et al. (2021) as clearly indicated in the text.

  1. There are some typos, e.g., (Line 287) "secondary" instead of "secondary" in the algorithm.

Response 8: The typo was corrected

  1. There were no experiments done in this study. Not sure why there is a whole section of it? It is usually common for each developed mathematical model, and it needs to be verified and then validated. What in this section, to my best knowledge, is really validation, not verification.

Response 9:   We change verification to validation.

  1. (Line 300 & 301) the author mentioned they used a history matching of recovery factor. The authors need to address and explain this.

Response 10:  These parameters were estimated by Mozo et al. (2018) from a core-flooding test of nanofluids injection and a durability test at reservoir conditions reported by Zabala et al. (2016); the parameters were estimated by matching the experimental observations of the oil recovery curve before and after the injection, as well as the durability. The latter was included in the manuscript

  1. Most, if not all, of the figures, are lack explanation and interpretation. For example, the authors just put figure 1 from the previous study without giving details of the reason behind it and why they address it here. They apply to all figures and charts.

Response 11: We do not agree with the reviewer. All figures are introduced and analyzed, explained, and interpreted.  

Figure 1 and Figure 3 are input data to the model. Therefore, they do not need any explanation and interpretation.  

The reviewer states that the situation… “apply to all figures and charts”. The statement is vague and lacks professionalism. Please indicate clearly what figures are lack explanation and interpretation and why you consider it, for the authors to correct/clarify and enhance the quality of the paper.

  1. (Figure 2) instead of having a long figure title, it would be much easier if there is a legend.

Response 12: The authors prefer a long title when additional explanatory information is provided.   

  1. (Line 351) "The skin factor of well C1, before and after the intervention, was history matched." Please explain.

Response 13:  A common practice in reservoir engineering is to history-match unknown parameters. Skin factor is the only unknown in the pre-treatment stage. It can increase over time and it can be estimated using a well test. The well does not have a recent well test.  A reservoir simulator requires the skin factor. Therefore, the authors matched the oil and water production rates measurements to estimate the skin factor. The obtained skin factor was consistent with other wells in the oil field.

  1. (Line 397) "site 2 follows a different dynamic behaviour from site 1." What kind of different dynamic do the authors refer to, and why does this behaviour occur?

Response 14: The model assumes that nanoparticles attachment on active site 2 is reversible, while the attachment on active site 1 is irreversible. The irreversible attachment assumption of the two-sites model (Zhang, 2012) is consistent with many core-flooding experiments. Afekare et al. (2021), using atomic force microscopy.  observed that nanoparticles can irreversibly be adsorbed onto the solid matrix.

  1. Overall results section, this section has great results and promising work. However, it lacks interpretation and explanation of these results. For example, (Line (430)) "Then, a decline in the production rate is observed in Figure 9."

Response 15:  Please refer to Response 11. The authors acknowledge specific comments, questions, and suggestions. 

Please let the authors know the specific parts of the article that requires interpretation and explanation.

Please let the authors know what promising work can potentially be included in this manuscript.

The comment "Then, a decline in the production rate is observed in Figure 9" was clarified and explained in the new version of the manuscript. Please refer to the text in yellow in the first paragraph, pp. 20.

  1. Some references needed adjustment and corrections. Would you please review them carefully?

Response 16: References were adjusted and corrected.

Reviewer 3 Report

The manuscript “Mathematical modeling and experimental verification of nanoparticles injection in heavy hydrocarbon reservoirs for wettability alteration” represents the numerical model for nanoparticle interaction in heavy oil reservoirs during production. The paper is well written and organized. The following are my recommendations:

  1. Page 5 Line 132: The authors mentioned a previous model that is extended by Valencia et al. (2019), but they did not cite which model they were referring to.
  2. Page 5 Line 142: Authors stated that there is an improvement on the simulation time but they are not disclosing how much improvement they accomplish. This statement is rather vague.
  3. Page 5 Line 145: There is a typo in here. It has to be “linear and radial”
  4. Page 7 Line 196: Authors are stating that they assume miscibility occurs between oil and diesel but they did not disclose what is rate of miscibility. Are they fully miscible at room temperature and reservoir temperature? The assumption regarding the miscibility is not clear in the text.
  5. Can the authors confirm Eq.1? I guess some terms are missing in the equation: The porosity and the saturation terms in accumulation is missing as well as the wi,p term in source. Also, can the authors provide the equation for Darcy velocity they have used?
  6. Can the authors provide the saturation equation? How are you solving for saturation?
  7. Page 9, Line 241: What is the ε r ?
  8. Page 10, Line 266: What is the viscosity equation? Eq.12 is not a viscosity equation.
  9. Page 6: Governing equations should be rewritten. It is very chaotic and hard to follow. Can you provide which equations you used in a more explicit way.
  10. Page 11, Line 288: How do you solve for the primary variables? Algorithm 1 only shows how you implemented GMRES mathematically. You can explain step-by-step you solve for pressure first, then saturation explicitly and the concentration or if you are solving implicitly define which terms are solved implicitly which ones explicitly?
  11. Page 12 Table 2: On Zabala (2016) the diameter was reported as 3.75 cm, but on this manuscript, it is given as 3.5 cm. If it’s not a typo, what was the reason of this change while keeping all other parameters and the results the same?
  12. Page 12, can you provide the initial conditions of the system in the experiment?
  13. Page 13: Same for the reservoir scale, what is the initial conditions of the system?
  14. Authors are mentioning wettability alteration in the introduction and conclusion as the effect of nanoparticles but they are not stating the formation type, mineralogy or initial wettability of these reservoirs. Can authors elaborate on these for the readers have a better understanding of the improvement on the reservoirs?

Author Response

Frist of all, we’d like to thank the reviewer for the comments and suggestions to improve the quality of this paper.

  1. Page 5 Line 132: The authors mentioned a previous model that is extended by Valencia et al. (2019), but they did not cite which model they were referring to.

Response 1: Agree. The model refers to the one presented by Mozo et al. (2018). A comment was included in the paragraph to present the main differences of these models. 

  1. Page 5 Line 142: Authors stated that there is an improvement on the simulation time but they are not disclosing how much improvement they accomplish. This statement is rather vague.

Response 2: The sentence is complemented by including that the model has better predictive capabilities and faster performance than the model presented by Valencia et al (2019).  The computational time was reduced in 25.9% in a 1D case and 78.2% in a 3D case

  1. Page 5 Line 145: There is a typo in here. It has to be “linear and radial”

Response 3: The typo was corrected

  1. Page 7 Line 196: Authors are stating that they assume miscibility occurs between oil and diesel but they did not disclose what is rate of miscibility. Are they fully miscible at room temperature and reservoir temperature? The assumption regarding the miscibility is not clear in the text.

Response 4:

It is expected that diesel and oil are fully miscible at both reservoir and room conditions. However, given the differences in viscosity, density, and composition, the mixing rate is finite. We assume that the mixing between diesel and the residual oil is instantaneous.

The following assumption was included

  • The carrier fluid (diesel and a nanoparticles dispersant) is miscible with oil at reservoir conditions and the mixing between residual oil and diesel is instantaneous

  1. Can the authors confirm Eq.1? I guess some terms are missing in the equation: The porosity and the saturation terms in accumulation is missing as well as the wi,p term in source. Also, can the authors provide the equation for Darcy velocity they have used?

Response 5: We confirm the validity equation 1.

The formulation presented here follows a compositional approach described by Echavarria et al. (2021). Particularly the black oil fluid model is incorporated following Wang (2007). In the (pseudo) compositional formulation, primary variables are pseudocomponent moles.  (Ni is the ith pseudo-component moles). Saturation is further calculated following Wang (2007) algorithm: as the moles of oil and gas pseudocomponents are the primary variables (please refer to Table 1) along with pressure, their distribution in the oleic and volatile phases are determined by the (pseudo) K-values (equilibrium constants) obtained from the pseudo-compositional approach Wang (2007). Having the densities of the phases, saturation can be computed straightforward. 

w_{i,p}, saturation and porosity are implicitly included in the accumulation term

N_{i} =  ­ \sum _p {w_{i,p} W_p \phi \S_p}

Darcy equation is included in the text.

  1. Can the authors provide the saturation equation? How are you solving for saturation?

Response 6: Please refer to response 5.

  1. Page 9, Line 241: What is the ε r ?

Response 7: is the rock volume fraction. The definition was included in the manuscript.

  1. Page 10, Line 266: What is the viscosity equation? Eq.12 is not a viscosity equation.

Response 8: Equation (12) was included to denote that viscosity is dependent on the concentration of nanoparticles, temperature, and pressure. A comment was included.

  1. Page 6: Governing equations should be rewritten. It is very chaotic and hard to follow. Can you provide which equations you used in a more explicit way.

Response 9: We do not agree with the reviewer. Here, a (pseudo) compositional formulation is presented, not a black oil model. This is one of the contributions of this work. We included equation (1.1) and (1.2)

The elements to understand the proposed formulation are found in the text:

  • The statement that the model is structured following a compositional formulation.
  • The algorithm that translates the PVT information (Wang 2007).

To avoid confusion, a sentence is included indicating that the model is solved for pressure and amount of matter in the last paragraph of pp. 11.

  1. Page 11, Line 288: How do you solve for the primary variables? Algorithm 1 only shows how you implemented GMRES mathematically. You can explain step-by-step you solve for pressure first, then saturation explicitly and the concentration or if you are solving implicitly define which terms are solved implicitly which ones explicitly?

Response 10: The solver is fully implicit for the primary variables. The step by step was included in the last paragraph of pp. 11.

  1. Page 12 Table 2: On Zabala (2016) the diameter was reported as 3.75 cm, but on this manuscript, it is given as 3.5 cm. If it’s not a typo, what was the reason of this change while keeping all other parameters and the results the same?

Response 11: The typo was corrected.

  1. Page 12, can you provide the initial conditions of the system in the experiment?

Response 12: The initial conditions of the system before the displacement test were added to Table 2.

  1. Page 13: Same for the reservoir scale, what is the initial conditions of the system?

Response 13: The initial conditions of the reservoir case are already reported in Table 3.

  1. Authors are mentioning wettability alteration in the introduction and conclusion as the effect of nanoparticles but they are not stating the formation type, mineralogy or initial wettability of these reservoirs. Can authors elaborate on these for the readers have a better understanding of the improvement on the reservoirs?

Response 14. We included more information about the reservoirs in the first paragraph of pp. 15. 

Round 2

Reviewer 2 Report

Accept with the current form.

No more comments are required at this stage as the authors addressed all the comments probably. 

Reviewer 3 Report

The authors revised the manuscript appropriately. Their explanations to the raised questions suffice.